# UReader: Universal OCR-free Visually-situated Language Understanding with Multimodal Large Language Model

**Jiabo Ye**[1*], **Anwen Hu**[2*], **Haiyang Xu**[2†], **Qinghao Ye**[2],
**Ming Yan**[2†], **Guohai Xu**[2], **Chenliang Li**[2], **Junfeng Tian**[2], **Qi Qian**[2], **Ji Zhang**[2],
**Qin Jin**[3], **Liang He**[1], **Xin Lin**[1], **Fei Huang**[2]

[1]East China Normal University
[2]DAMO Academy, Alibaba Group    [3]Renmin University of China

jiabo.ye@stu.ecnu.edu.cn    {huanwen.haw,ym119608,shuofeng.xhy}@alibaba-inc.com

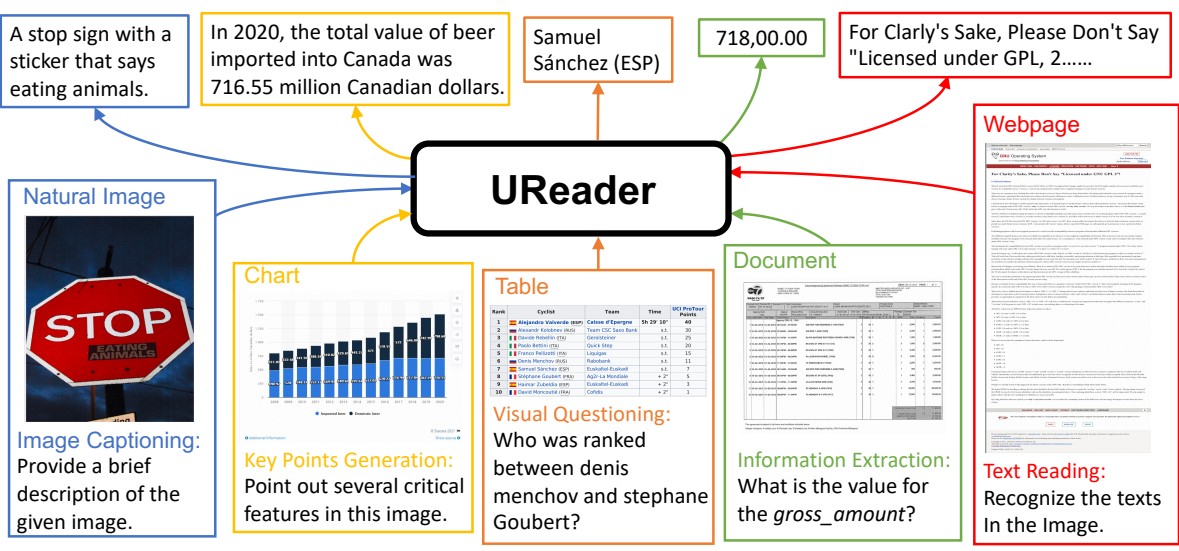

Figure 1: The OCR-free Visually-situated Language Understanding performance of UReader on various types of images and tasks.

## Abstract

Text is ubiquitous in our visual world, conveying crucial information, such as in documents, websites, and everyday photographs. In this work, we propose UReader, a first exploration of universal OCR-free visually-situated language understanding based on the Multimodal Large Language Model (MLLM). By leveraging the shallow text recognition ability of the MLLM, we only finetuned 1.2% parameters and the training cost is much lower than previous work following domain-specific pretraining and finetuning paradigms. Concretely, UReader is jointly finetuned on a wide range of Visually-situated Language Understanding tasks via a unified instruction format. To enhance the visual text and semantic understanding, we further apply two auxiliary tasks with the same format, namely text reading and key points generation tasks. We design a shape-adaptive cropping module before the encoder-decoder architecture of MLLM to leverage the frozen low-resolution vision encoder for processing high-resolution images. Without downstream finetuning, our single model achieves state-of-the-art ocr-free performance in 8 out of 10 visually-situated language understanding tasks, across 5 domains: documents, tables, charts, natural images, and webpage screenshots. Codes and instruction-tuning datasets are released at https://github.com/LukeForeverYoung/UReader.

## 1 Introduction

Leveraging strong Large Language Models as the language decoder, some recent works propose Multimodal Large Language Models (MLLMs) (Zhu et al., 2023; Liu et al., 2023a; Ye et al., 2023; Li et al., 2023) and achieve promising vision-and-language understanding performance. Surprisingly, without in-domain training, these MLLMs exhibit shallow zero-shot visual text recognition ability when fed a low-resolution image with salient text information (Ye et al., 2023; Liu et al., 2023b). However, due to the variety of image types and the wide range of image sizes, they are still far from universal visually-situated language understanding, such as extracting information from documents, reading texts from webpages, and visual question and answering on tables, as shown in Figure 1.

---

* Equal contribution
† Corresponding authors

Existing works for visually-situated language understanding can be categorized into two-stage (Xu et al., 2021; Huang et al., 2022; Yang et al., 2021) and end-to-end (Davis et al., 2022; Kim et al., 2022; Lee et al., 2022) methods according to whether relying on an off-the-shelf OCR model or API. These works all follow a domain-specific pretraining and finetuning paradigm, thus leading to high training costs, e.g. end-to-end model Donut (Kim et al., 2022) costs more than 192 A100 days.

Inspired by the shallow text recognition ability of existing MLLMs, in this work, we propose **UReader** for universal OCR-free visually-situated language understanding, which leverages the multimodal Large Language Model via low-cost instruction tuning (Dai et al., 2023). Different from previous works, we forgo pretraining tasks by leveraging the existing MLLM and directly finetune MLLM by taking full advantage of various Visually-situated Language Understanding datasets. To make the most of the strong language understanding ability of MLLM, we convert all tasks into the vision-language instruction tuning format. Besides, to enhance text recognition and semantic understanding ability across diverse domains, we design auxiliary text reading and key points generation tasks in the same instruction format. To utilize the low-resolution encoder of MLLM for processing high-resolution images and avoid blurry and distortion problems due to resizing, we propose a shape-adaptive cropping module to cut a high-resolution image into multiple local images. Each image is firstly independently encoded with the frozen visual encoder and a trainable visual abstractor and then concatenated to feed into the language decoder. Moreover, we add learnable crop position encoding to help the model correlate local images and add a resized global image to alleviate salient information loss due to cropping.

Our contributions in this work are four-fold:
- We first propose instruction tuning with Multimodal Large Language Models for OCR-free Visually-situated Language Understanding.
- We build an instruction-tuning dataset covering 5 domains of visually-situated language understanding: document, table, chart, natural image, and webpage screenshot.
- We design a shape-adaptive cropping module to utilize the frozen low-resolution vision encoder for processing high-resolution images.
- UReader achieves state-of-the-art OCR-free

performance in 8 out of 10 tasks, across 5 domains.

## 2 Related Work

**Visually-situated Language Understanding** aims to comprehend images containing rich text information. The image types are quite diverse, covering document (Mathew et al., 2021, 2022; Stanislawek et al., 2021; Svetlichnaya, 2020; Zhang et al., 2023), table (Pasupat and Liang, 2015; Chen et al., 2020), chart (Masry et al., 2022; Methani et al., 2020; Kafle et al., 2018; Kahou et al., 2018), natural image (Singh et al., 2019; Mishra et al., 2019; Biten et al., 2019; Hu et al., 2021), webpage screenshot (Tanaka et al., 2021; Chen et al., 2021), etc. Tasks of Visually-situated Language Understanding range from visual question answering, image captioning, information extraction to natural language inference.

According to whether using off-the-shelf OCR models or APIs to recognize texts from images, existing work can be divided into two-stage models (Xu et al., 2021; Huang et al., 2022; Tang et al., 2023; Yang et al., 2021) and end-to-end models (Kim et al., 2022; Davis et al., 2022; Lee et al., 2022). Two-stage work always designs pretrianing tasks to learn cross-modality alignment between visual inputs and text inputs. For example, for document understanding, UDOP (Tang et al., 2023) design a Joint Text-Layout Reconstruction task to recover masked texts and layout information given the visual inputs and retained text inputs. LayoutLMv3 (Huang et al., 2022) applies a Masked Image Modeling task to recover masked image tokens with the context of their surrounding text and image tokens. Without the help of an off-the-shelf OCR model, end-to-end models need to learn text recognition with a high-resolution image encoder during the pretraining stage. For example, Pix2Struct (Lee et al., 2022) proposes a Screenshot Parsing pretraining task, where the model needs to generate the complete HTML DOM tree with only a masked webpage screenshot as the input. Donut (Kim et al., 2022) designs a pretraining task to generate all texts in the document image. These work all follow a domain-specific pretraining and finetuning paradigm and therefore ask for high training costs, e.g. Donut is trained for more than 192 A100 days. In this work, by leveraging the shallow text recognition ability of Multimodal Large Language Models, we propose to directly perform

instruction tuning across various types of images and greatly reduce the training cost for universal visually-situated Language Understanding.

**Multimodal Large Language Model** is developed to empower the Large Language Model with multi-modality understanding ability, especially for vision information. These work (Huang et al., 2023; Zhu et al., 2023; Liu et al., 2023a; Ye et al., 2023; Li et al., 2023; Dai et al., 2023) mainly connect a pre-trained vision encoder (usually CLIP VIT-L/14 (Radford et al., 2021)) with a strong large language model, such as LLaMA (Touvron et al., 2023). These MLLMs show some emergent abilities, including shallow zero-shot text recognition ability (Liu et al., 2023b). However, they are still far from universal visually-situated language understanding. Firstly, due to the pretraining data for the vision encoder being mostly natural images, MLLMs show barely acceptable text understanding performance on natural images but bad performance on other types, such as document (Liu et al., 2023b). Secondly, most images for visuall-situated language understanding are high-resolution. Rescaling them to low resolution to adapt to the vision encoder can cause the texts blurry and distorted. In this work, we propose to fully leverage the shallow text recognition ability of MLLMs and perform instruction tuning to enhance its universal understanding ability across 5 domains. Besides, we design a shape-adaptive cropping module to alleviate the text blur and distortion problem.

## 3 UReader

The primary goal of UReader is to efficiently utilize existing MLLMs for Visually-situated Language Understanding tasks. In this work, we utilize but are not limited to, the mPLUG-Owl (Ye et al., 2023) as our basic MLLM. Figure 2 presents an overall architecture of UReader. The input image is firstly pre-processed by a shape-adaptive cropping module (in Section 3.1). The resulting sub-images are then simultaneously passed through the visual encoder and visual abstractor. To enable the large language model to correlate multiple cropped sub-images, we apply a crop position encoding module to introduce spatial information across sub-images. (in Section 3.2).

### 3.1 Shape-Adaptive Cropping Module

Images with texts have various aspect ratios and a great range of resolutions. Simply resizing the im-

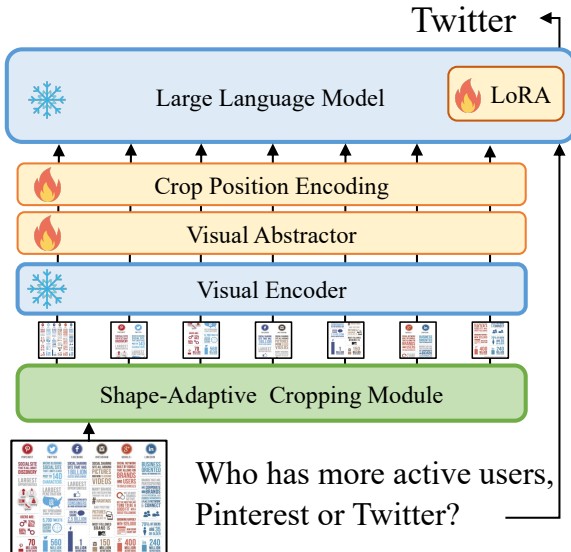

Figure 2: The overall architecture of UReader.

age to $H_v, W_v$ (raw resolution of the MLLM) can result in text being blurred, distorted, and unrecognizable. Thus we propose a shape-adaptive cropping module. Specifically, as shown in Figure 3, we pre-define grids $\{g = (n_h \times n_w)|n_h \cdot n_w \leq N_c, n_h \in \mathbb{N}, n_w \in \mathbb{N}\}$ with various shapes, where $n_h$ and $n_w$ denote the number of rows and columns of the grid $g$ and $N_c$ denotes the maximum number of the cells (sub-images). To select a suitable grid for an image $I$ with shape $H \times W$, two rules should be followed: (1) The grid should preserve the resolution of the image as much as possible, and (2) the grid should fit the aspect ratio of the input image. To measure the resolution coherence and shape similarity between the image and each grid, we calculate the resolution-related and resolution-agnostic insection over union $S_{rr}$ and $S_{ra}$ as follows:

$$S_{rr}(I, g) = \text{IoU}\left((H, W), (n_h H_v, n_w W_v)\right)$$
$$S_{ra}(I, g) = \text{IoU}\left((\frac{n_w H}{W}, n_w), (n_h, n_w)\right) \quad (1)$$

where IoU denotes the insection over the union between two rectangles centered and aligned with each other. The matched grid is selected by maximizing the matching score:

$$g^* = \arg\max_g S_{ra}(I, g) + S_{rr}(I, g) \quad (2)$$

where $g^*$ is the selected grid. Then, we resize the input image to $(n_h H_v, n_w W_v)$ and crop it to $n_h \cdot n_w$ local images. To maintain the global structure information of the image, we also resize the input

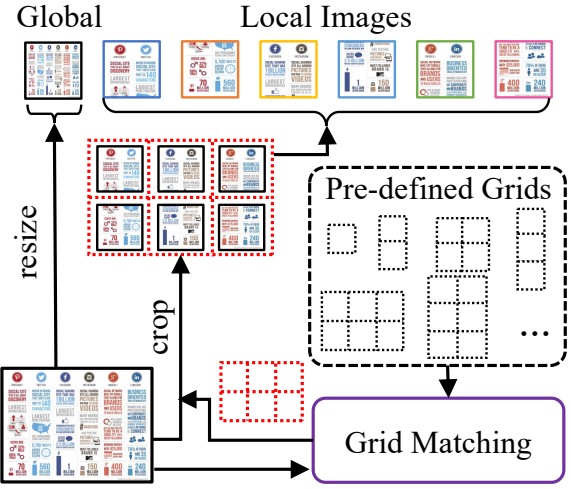

Figure 3: The Shape-Adaptive Cropping Module.

image to $(H_v, W_v)$ as a global image. All images are then passed on to the visual encoder and visual abstractor in parallel.

The visual encoder extracts visual feature $V \in \mathbb{R}^{N \times (H' \cdot W') \times d_v}$ from the input images $\mathbf{I} \in \mathbb{R}^{N \times H \times W \times 3}$, where $N = (n_h \cdot n_w) + 1$, $H' \cdot W'$ and $d_v$ denote the number and dimension of the extracted visual features, respectively. The visual abstractor further summarizes visual information and obtains higher semantic visual representations $V^l \in \mathbb{R}^{N \times N_q \times d_l}$ in language feature space by several learnable queries, where $d_l$ denotes the dimension of language feature space and $N_q$ denotes the number of learnable queries.

## 3.2 Cropped Images Modeling with LLM

MLLMs are mostly trained with a single image as the input. Due to the cropping module, we need to input visual features from multiple images into the language model. The 1-dimensional position embeddings of LLM can not reflect the spatial position of each sub-image, which is critical to correlate local images. Therefore, we incorporate a 2-dimensional crop position encoding to help the language model to understand the spatial relationship between cropped images. Specifically, we assign a location index $(i, j)$ for each cell of the selected grid and obtain their row embedding and column embedding by two auxiliary embedding layers as follows:

$$\mathbf{e}_{i,j}^{row} = \text{Embedding}_{\text{row}}(i)$$
$$\mathbf{e}_{i,j}^{column} = \text{Embedding}_{\text{column}}(j) \quad (3)$$
$$\mathbf{e}_{i,j} = \mathbf{e}_{i,j}^{row} + \mathbf{e}_{i,j}^{column}$$

where $\mathbf{e}_{i,j} \in \mathbb{R}^{D_l}$ denotes the crop position embedding of the cell $(c_i, c_j)$. We add the embedding to the visual feature of each cell in the language space via broadcasting along the dimension of learnable queries: $\bar{V}_{i,j}^l = V_{i,j}^l + \mathbf{e}_{i,j}$. We then reshape the visual features into $\bar{\mathbf{V}}^l \in \mathbb{R}^{(N \cdot N_q) \times d_l}$. The resulting spatial-aware visual features and word embeddings of the input sentences are concatenated at sequence dimension and sent to the large language model.

In order to enhance the language model's ability to effectively model multiple images while keeping low training costs, we freeze the origin language model and adopt the low-rank adaptation approach (LoRA) (Hu et al., 2022).

## 4 Instruction Tuning

For developing a universal visually-situated language understanding model that could process various types of images and perform different comprehension tasks, we conduct low-cost instruction tuning with a Multimodal Large Language Model. Without introducing any large-scale pretraining datasets, we directly ensemble multiple downstream datasets and perform joint training. Different downstream tasks are all reorganized to the unified instruction format (Dai et al., 2023). Besides, we design auxiliary text reading and key points generation tasks to enhance text recognition and semantic understanding ability.

### 4.1 Tuning Tasks

**Unified downstream task.** Downstream tasks of Visuall-situated Language Understanding cover Visual Question Answering, Information Extraction, Natural Language Inference, and Image Captioning. For developing a universal model, we reorganize all tasks into the instruction tuning format (Dai et al., 2023). Concretely, for the Visual Question Answering task, the question is directly used as the instruction: "Human: {question} AI: {answer}". For the Information Extraction task, each category and value pair is expressed with a prompt as "Human: What is the value for the {category}? AI: {value}". If some categories don't exist in the image, the value is 'None'. In the raw annotation of the Natural Language Inference task, '1' means 'Entailed' and '0' means 'Refuted'. We reorganize the NLI task by constructing the instruction "Human: {statement}, Yes or No? AI: {answer}", where 'Yes' means 'Entailed'. For the Image captioning task, we refer to 11 prompts from LLaVa

(Liu et al., 2023a) to instruct the model to briefly describe the image and randomly choose 1 prompt for each caption, such as "Human: Provide a brief description of the given image. AI: {caption}".

**Text Reading task.** Text Recognition is a basic ability for OCR-free Visuall-situated Language Understanding. Therefore, we apply an auxiliary Text Reading task to strengthen text recognition ability across different domains. With the text and position information in the image, we organize the texts in the common reading order: from top to down, from left to right. Directly utilizing all texts as targets (Kim et al., 2022) will result in the model focusing on generating the starting texts and neglecting others to reduce the loss. Instead, we randomly choose a split position $p$ from $\{0, \frac{L}{6}, \frac{2L}{6}, ..., \frac{5L}{6}\}$, where $L$ is the text sequence length. The left part is used as the input and the right one is the target. $p = 0$ means to generate all texts while other cases ask the model to continue reading following the input texts. Such a design could enforce the model to read different parts of texts with the context. Starting texts always convey key information about the image, such as the chart title. Therefore, we apply a bigger sample rate (0.5) for the '0' position and 0.1 for other positions. To distinguish reading from the beginning and continuing reading, we design two groups of prompts and randomly choose 1 prompt for each sample. For example, an instruction of reading from the beginning can be "Human: Recognize text in the image. AI: {all texts}" and an instruction of continuing reading can be "Human: The words on this picture are {left texts}. Continue reading the text. AI: {right texts}".

**Key Points Generation task.** Large Language Models learn strong understanding ability from the tough language modeling task. Therefore, for stronger vision-and-language semantic comprehension ability, we propose to design an auxiliary Key Points Generation task, which requires the model to give some key points about the image. To support this task, we collect QA pairs of each image and convert them to declarative sentences with Vicuna (Vicuna, 2023). These declarative sentences are finally regarded as key points about the image. We also build a set of templates to instruct this task, such as "Human: Identify some key points in this picture. AI: {key points}".

All templates for Text Reading and Key Points Generation tasks can be found in Appendix D.

## 4.2 Instruction Data Resources

**Document.** DocVQA (Mathew et al., 2021) comprises 50k question and answer(QA) paris on 12k document images from UCSF Industry Documents Library. InfographicsVQA (InfoVQA) (Mathew et al., 2022) collects 5k diverse infographics from the internet and annotates 30k QA pairs. DeepForm[*][1] (Svetlichnaya, 2020) and Kleister Charity (KLC) (Stanislawek et al., 2021) are two Information Extraction datasets. DeepForm[*] contains 1.1k documents related to election spending. 2.7k documents of KLC come from published reports of charity organizations.

**Table.** WikiTableQuestions (WTQ[*]) (Pasupat and Liang, 2015) comprises 2.1k table images from Wikipedia and is annotated with 23k question and answer pairs demanding comparison and arithmetic operations. TabFact[*] (Chen et al., 2020) is a Natural Language Inference dataset, which contains 112k 'entailed' or 'refuted' statements about 16k Wikipedia tables.

**Chart.** ChartQA (Masry et al., 2022) collects various topics and types of charts from four sources: Statista (statista.com), The Pew research (pewresearch.org), OWID (ourworldindata.org) and OECD (oecd.org). It totally contains 21k chart images and 32k QA pairs.

**Natural Images.** TextVQA (Singh et al., 2019) filters 28k natural images with texts from Open Images V3 (Krasin et al., 2017) and annotates 45k QA pairs. To support image captioning with reading comprehension, TextCaps (Sidorov et al., 2020) further collects 145k captions based on TextVQA.

**WebPage Screenshot.** VisualMRC (Tanaka et al., 2021) collects 5k full screenshots of webpages from 35 websites. There are 30k annotated QA pairs where answers are expressed in fluent sentences (avg. 9.53 words) and much longer than the ones of QA datasets mentioned above.

## 5 Experiments

### 5.1 Implementation Details

We conduct experiments on a recently proposed MLLM named mPLUG-Owl (Ye et al., 2023) without modifying its hyperparameters. The number of learnable queries of visual abstractor is 65. The dimension of hidden states $d_v$ and $d_l$ are 1024. For the shape-adaptive cropping module, we set the

---

[1]Superscript [*] means the reformulated or modified version in DUE-benchmark (Borchmann et al., 2021)

Table 1: Comparison with ocr-free methods on various types of visually-situated language understanding tasks. 'TSFT' means task-spcific fine-tuning on the downstream dataset. 'underline' means achieving 80% SOTA performance.

| Model | Train Param | TS FT | Doc VQA | Info VQA | Deep Form | KLC | WTQ | TabFact | ChartQA | TextVQA | TextCaps | Visual MRC |
|---|---|---|---|---|---|---|---|---|---|---|---|---|
| Dessurt | 127M | ✓ | 63.2 | - | - | - | - | - | - | - | - | - |
| Donut | 176M | ✓ | 67.5 | 11.6 | 61.6 | 30.0 | 18.8 | 54.6 | 41.8 | 43.5 | 74.4 | 93.91 |
| Pix2Struct$_{base}$ | 282M | ✓ | 72.1 | 38.2 | - | - | - | - | 56.0 | - | 88.0 | - |
| Pix2Struct$_{large}$ | 1.3B | ✓ | **76.6** | 40.0 | - | - | - | - | 58.6 | - | 95.5 | - |
| UReader | 86M | ✗ | 65.4 | **42.2** | 49.5 | **32.8** | **29.4** | **67.6** | **59.3** | **57.6** | **118.4** | **221.7** |

maximum number of cells $N_c$ to 20 by default. During instruction tuning, the maximum sequence length is limited to 2048, and $H_v, W_v$ are set to 224 to match the pretrained resolution of the visual encoder. For LoRA, we set the rank $r = 8$. The learning rate schedule uses a linear warmup of 36 steps to $1e^{-4}$, followed by cosine decay to 0. The batch size is set to 256. For better convergence of each dataset, DocVQA is up-sampled 3 times, InfoVQA, WTQ, DeepForm, and KLC are up-sampled 2 times. The instruction tuning process takes 16 A100 days for 20k training steps (10 epochs).

## 5.2 Evaluation

We use official training splits as tuning data and evaluate models on test splits. Following previous works (Borchmann et al., 2021; Lee et al., 2022), DocVQA and InfoVQA are evaluated by ANLS (Biten et al., 2019), DeepForm and KLC are evaluated by F1 score. WTQ, TabFact and TextVQA are evaluated by accuracy. ChartQA is evaluated with the relaxed accuracy (Methani et al., 2020). TextCaps and VisualMRC are measured by CIDEr (Vedantam et al., 2015). Evaluation of TextVQA and TextCaps are performed with the official challenge website.

## 5.3 Main Results

We first compare UReader with state-of-the-art ocr-free models on 10 datasets. For fair and consistent comparison across all datasets, we finetune the strong and accessible baseline Dount on unreported datasets. As shown in Table 1, UReader achieves state-of-the-art performance in 8 tasks across 5 domains, covering Visual Question Answering, Information Extraction, Natural Language Inference and Image Captioning tasks. With much fewer trainable parameters (86M vs 1.3B) and without a specific finetuning stage, UReader outperforms

the strong pretriaining model Pix2Struct$_{large}$ in InfoVQA, ChartQA, and TextCaps. Considering that Pix2Struct$_{large}$ is trained more than 170k steps with a batch size of 1024 on 128 TPUs, this validates that with the help of open-domain Multimodal Large Language Models, learning costs for universal visually-situated language understanding can be greatly reduced. More detailed analysis can be found in Appendix B.

## 5.4 Ablation Study

We perform comprehensive ablation experiments to validate the contribution of two auxiliary tasks, trainable architectures, cross-domain joint training and the design of shape-adaptive cropping module.

**Auxiliary Tasks.** As shown in Table 2, dropping the Key Points Generation task (r10 vs r2) causes a performance decrease on all domains of datasets, demonstrating that this task helps the model better understand the vision-and-language semantic. Further removing the Text Reading task (r2 vs r1) causes more significant performance degradation, which validates the importance of enhancing text recognition ability across different domains.

**Trainable Architectures.** Both the visual abstractor and LoRA in LLM are finetuned in UReader (r10). Freezing either the visual abstractor (r3) or LoRA (r4) causes performance decrease, which demonstrates that both the vision and language parts should be finetuned for adjusting to Visually-situated Language Understanding.

**Cross-domain Joint Training.** After removing 4 document datasets from the training data, UReader achieves worse performance (r10 vs r5) on the table, natural image, and webpage domains, validating that images of different domains share some common characteristics and cross-domain joint training improves the universal performance. Besides, although trained without document data,

Table 2: Ablation study about auxiliary training tasks, trainable model architectures, cross-domain joint training and shape-adaptive cropping. 'KPG' and 'TR' refer to Key Points Generation and Text Reading tasks, respectively. 'Abs' refers to the visual abstractor. 'Doc Data' means using 4 document datasets as training data or not. 'Global' means using a resized global image as input. 'Crops' refers to $N_c$, the maximum number of local images after cropping. 'CropPos' refers to the crop position embedding.

| | Tasks | | Trainable | | Doc Data | Shape-adaptive Cropping | | | DocVQA | WTQ | ChartQA | TextVQA | Visual MRC |
| | KPG | TR | Abs | LoRA | | Global | CropPos | Crops | | | | | |
|---|---|---|---|---|---|---|---|---|---|---|---|---|---|
| r1 | | | ✓ | ✓ | ✓ | ✓ | ✓ | 20 | 56.7 | 22.9 | 56.7 | 54.3 | 205.0 |
| r2 | | ✓ | ✓ | ✓ | ✓ | ✓ | ✓ | 20 | 64.3 | 28.1 | 58.6 | 56.0 | 213.5 |
| r3 | ✓ | ✓ | | ✓ | ✓ | ✓ | ✓ | 20 | 52.4 | 20.5 | 43.5 | 54.9 | 194.9 |
| r4 | ✓ | ✓ | ✓ | | ✓ | ✓ | ✓ | 20 | 59.5 | 23.5 | 58.5 | 53.3 | 177.0 |
| r5 | ✓ | ✓ | ✓ | ✓ | | ✓ | ✓ | 20 | 46.2 | 27.4 | 59.8 | 54.0 | 185.6 |
| r6 | ✓ | ✓ | ✓ | ✓ | ✓ | ✓ | | - | 22.0 | 13.4 | 24.2 | 34.4 | 157.4 |
| r7 | ✓ | ✓ | ✓ | ✓ | ✓ | | ✓ | 9 | 58.0 | 24.7 | 58.9 | 55.5 | 215.3 |
| r8 | ✓ | ✓ | ✓ | ✓ | ✓ | | ✓ | 20 | 64.1 | 27.6 | **60.7** | 56.5 | 210.7 |
| r9 | ✓ | ✓ | ✓ | ✓ | ✓ | ✓ | | 20 | 62.8 | 26.7 | 58.7 | 55.4 | 181.1 |
| r10 | ✓ | ✓ | ✓ | ✓ | ✓ | ✓ | ✓ | 20 | **65.4** | **29.4** | 59.3 | **57.6** | **221.7** |

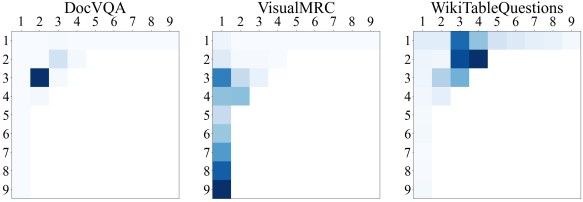

Figure 4: Visualization of the frequency of selected grid with shape-adaptive cropping module. The cell at row $i$ and column $j$ denotes the selected frequency of grid $(n_h = i, n_w = j)$. Deeper colors represent higher selection frequencies.

our model achieves a 46.2 score on the DocVQA dataset, showing the potential out-of-domain understanding ability of our training paradigm.

**Shape-adaptive Cropping.** The r6 in Table 2 represents directly tuning the mPLUG-Owl without any model revisions. With the shape-adaptive cropping, UReader achieves significantly better performance (r7 vs r6), showing that our cropping module is indispensable to leverage pretrained low-resolution vision encoder for universal visually-situated language understanding. Besides, increasing the cropping numbers (r8 vs r7) improves the model's performance. Due to the resolution of each local image being constant (224x224), more crops mean higher overall resolution and therefore achieves better performance. Furthermore, adding a resized global image bring a slight improvement in most datasets (r10 vs r8), validating that a complete image could alleviate possible information loss due to image cropping. Finally, dropping crop position encoding also hurts the model's perfor-

mance (r10 vs r9), proving the effectiveness of crop position encoding for correlating local images.

For alleviating the distortion problem due to resizing, we propose to crop images according to their raw aspect ratio. Figure 4 shows the frequency distribution of grids selected by our shape-adaptive cropping module on DocVQA, VisualMRC and WikiTableQuestions (the distribution on more datasets can be found in the Appendix A). For aesthetic purposes, we present the distribution with $N_c = 9$. Apparently, different domains of images have different shape distributions. For most document images in DocVQA, their height is greater than the width, while table images are the opposite. As webpages are scrollable, their screenshots are always in the form of a long rectangular shape. With the shape-adaptive cropping design, our model can easily adapt to various image shapes without domain-specific fine-tuning.

Text distortion may pose little influence on visual question answering because they are always about partial text information. But it is harmful for reading texts in the image because every text matters. For quantitative analysis of the influence of shape-adaptive design, we directly evaluate the performance of reading all texts. We choose the Bleu (Papineni et al., 2002) as the metric because it directly measures the n-gram overlap between the ground-truth and predicted text sequence. The evaluation set is built by combining 100 randomly-selected test images from each dataset. As shown in Table 3, compared with cropping all images with a fixed grid, UReader could better recognize texts

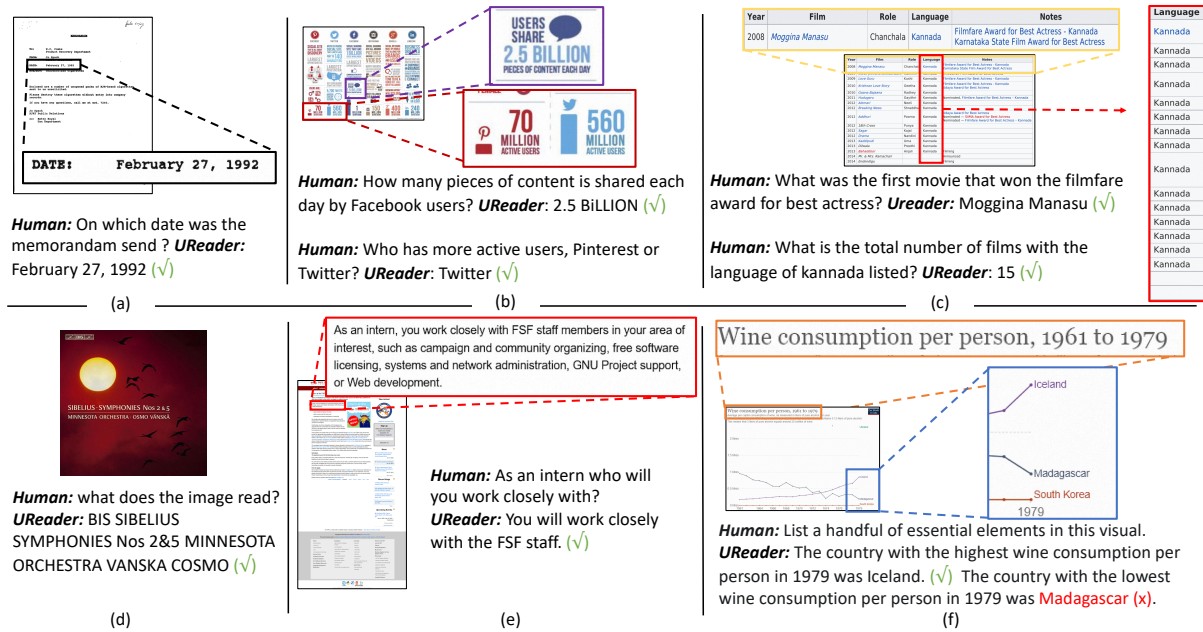

Figure 5: Qualitative results of UReader. Crucial regions are enlarged for clearer visualization.

Table 3: The Text Reading performance of UReader under the condition of $N_c = 9$. 'w/o adapt means removing the shape-adaptive design and cropping the image with a fixed grid $3 \times 3$.

| Model | Bleu1 | Bleu2 | Bleu3 | Bleu4 |
|---|---|---|---|---|
| UReader w/o adapt | 21.4 | 15.4 | 12.0 | 9.7 |
| UReader | **24.9** | **18.1** | **14.3** | **11.7** |

in the image due to our shape-adaptive design that alleviates the text distortion problem.

## 5.5 Qualitative Results

Figure 5 show some qualitative results produced by our UReader on different types of images. UReader could not only extract information from the document (case a), but also understand different instructions and provide corresponding answers by attending to different regions (case b). Table understanding always involves layout comprehension and statistics. As shown in case c, given a table image, UReader could well relate different columns to answer the 'first movie' and perform simple statistics about the 'total number'. As for images with multiple paragraphs of text, e.g. webpage screenshot in case e, UReader could also locate the relevant paragraph, understand the texts and answer the question accurately. Case d shows the text reading performance. With the help of the Text Reading task, UReader is able to read texts from top left to bottom right. But, due to the language decoding manner, when given an image with rich texts, such as a page of a book, the model often reads the beginning texts and then continues writing without watching the image. More qualitative results can be found in Appendix C. Finally, as shown in case f, UReader is able to list some key points about the chart by combining the title and line information. Listing key points in this work is just a superficial attempt at open-ended generation, and its performance is far from promising, e.g., UReader makes a mistake about the lowest line. More effort is needed towards a comprehensive understanding of images with rich text.

## 6 Conclusion

We first propose to leverage existing Multimodal Large Language Models for universal ocr-free visually-situated language understanding through low-cost instruction tuning. All downstream tasks are reorganized into a unified instruction-tuning format. Besides, we design the Text Reading task and Key Points Generation task to enhance text recognition and vision-and-language semantic comprehension abilities. To utilize the pre-trained vision encoder for processing high-resolution images, we design a shape-adaptive cropping module, which cuts the image into multiple local images considering its raw aspect ratio and resolution. UReader achieve state-of-the-art ocr-free performance in 8 out of 10 datasets, ranging from documents, tables, charts, and natural images to webpage screenshots.

## Limitations

Our experiments validate that UReader is able to correlate local images after cropping a high-resolution image. However, UReader struggles to understand multi-page documents (e.g. books and papers) due to lacking ability to correlate different pages and the limited sequence length of the decoder. Besides, UReader feeds an equal number of features for each local image into the language decoder. But, not all local images contain rich vision or text information. In the future, we will explore a more efficient way to encode different crops. Furthermore, the open-ended generation about Visually-situated Language understanding is far from well studied. We try developing key points generation ability in this work but more difficult generation tasks are not currently considered, such as giving the chain-of-the-thought of the answer. How to simulate such abilities through instruction tuning is a topic worth studying. Finally, the Text Reading task helps the model recognize texts, but the text reading performance with the LLM as the decoder is far from satisfactory due to the hallucination problem. Instructing the LLM to read texts strictly according to images is a challenging topic.

## Ethics Statement

Our UReader relies on multi-modal large language models that are trained on large-scale image and text data from the web and therefore may be subject to issues such as toxic language and bias (Bender et al., 2021). However, our model is further fine-tuned on publicly available datasets and is used specifically in the domain of visually-situated language understanding, where these issues have minimal impact.

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

## A  Grid Distribution on Downstream Datasets

We visualize the frequency distribution of grids selected by our shape-adaptive cropping module on all ten datasets in Figure 6. The wide variety of image shapes in downstream tasks highlights the crucial role of the shape-adaptive cropping module.

## B  Detailed Analysis on Performance

### B.1  Underperforms Ocr-Free Baselines on DocVQA and DeepForm

It can be seen that UReaderunderperforms ocr-free baselines on DocVQA and DeepForm. There are two main factors: (1) Donut performs the pretraining on large-scale document dataset IIT-CDIP (11M document images), which is the same domain as DocVQA and DeepForm. But UReader does no have a pretraining process and is just instruction finetuned on ensembled datasets (less than 0.5M assorted images). Training with more document images brings better performance. (2) The pretraining task of Pix2struct is to predict the HTML dom tree of a masked web screenshot, which requires the model to fully understand the layout information of the image. But UReader is trained to read texts from top to down, from left to right, which requires a weaker layout understanding ability. The pretraining on layout understanding also leads to improved performance on DocVQA.

The conclusion can also be substantiated by the observations on the other two datasets (i.e., InfoVQA and KLC) included in the document domain as previous work (Tang et al., 2023). For the InfoVQA dataset, the image is poster style and the layout is not as important as DocVQA and DeepForm but the relationship between text and vision objects matters more, like natural image and chart image. As for the KLC dataset, ocr-free models are only fed with the first page (always the cover of a report) , where the layout is much simpler than DocVQA and DeepForm. Therefore, UReadercan outperform baselines on these two document datasets.

In summary, compared with ocr-free model Donut and Pix2Struct, due to the pretrianing of MLMM on open-domain datasets, UReaderis better at understanding cross-modality relationships in the image but weaker at comprehending text layout information without large-scale document pretraining and specific layout understanding tasks.

### B.2  Compared with Pipeline Methods

We list the performance of state-of-the-art pipeline models in Table 4. We can summarize from the results that there are two distinct aspects. Firstly, our model achieves comparable or slightly worse results compared to the pipeline methods on TextVQA, ChartQA, InfoVQA, TextCaps and TabFact. Secondly, there is a obvious gap between our model and pipeline methods on DocVQA, DeepForm, KLC, WTQ and VisualMRC.

For the first aspect, there are two reasons for the similarity performance: (1) Modeling the diverse relationship between visual objects and text presents challenges for both pipeline-based methods and OCR-free methods. TextVQA, TextCaps and InfoVQA requires the relation understanding between text and visual objects (i.e. logos, icons and common objects). ChartQA asks for trend comprehension of lines. Understanding such complex cross-modality relation is challenging for both ocr-free and pipeline methods. (2) The simplicity of task formats can reduces performance gaps. Tabfact is a simply binary classification task resulting the small performance gap.

For this second aspect, the main performance gap appears in three categories of datasets: document, table, and webpage screenshot. The reasons are two folds: (1) The gap in terms of text recognition and layout extraction. In document, table and website, text is the dominant information source and the layout(e.g. row and column layout in table) is relatively uniformer than the chart and natural images. Therefore, with pre-extracted texts and layout information, it is more easy to understand the image. But for OCR-Free models, such as our UReader and Donut, it's still challenging to fully recognize all texts. (2) The gap in terms of modeling capacity on multi-page document input. for multiple-page document datasets KLC (98% > 4 pages) and DeepForm (75% > 1 pages), OCR-Free models only input the first page and lose much information.

### B.3  Zero-shot Performance

We test the zero-shot performance of UReader on unseen dataset OCR-VQA. With the same evaluation metrics, UReader outperforms mPLUG-Owl (41.1 vs 28.6) and a recent work UniDoc (Feng et al., 2023) (41.1 vs 34.5) with the training of layout prediction. The results show that the zero-shot performance of our method on unseen domains is

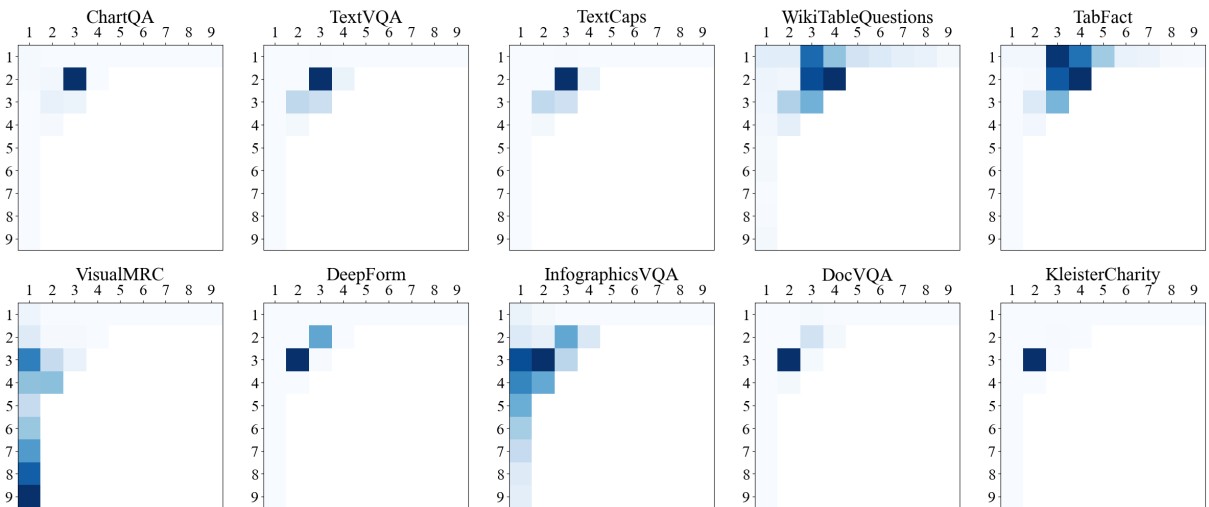

Figure 6: Visualization of the frequency of selected grid with the shape-adaptive cropping module on 10 downstream datasets.

| | DocVQA | InfoVQA | DeepForm | KLC | WTQ | TabFact | ChartQA | TextVQA | TextCaps | VisualMRC |
|---|---|---|---|---|---|---|---|---|---|---|
| OCR-Pipline | 84.7(UDOP) | 47.4(UDOP) | 85.5(UDOP) | 82.8(UDOP) | 47.2(UDOP) | 72.9(UDOP) | 70.5(DePlot) | 56.3(PreSTU) | 139.1 (PreSTU) | 364.2(LayoutT5) |
| UReader | 65.4 | 42.2 | 49.5 | 32.8 | 29.4 | 67.6 | 59.3 | 57.6 | 118.4 | 221.7 |

Table 4: Performance comparison between UReaderand state-of-the-art pipeline methods.

acceptable.

## C   More Qualitative Results

### C.1   Downstream Results

More qualitative results on natural images, charts, tables, documents and webpage screenshots are shown in Figure 7-11.

Figure 11 show a sample of Text Reading and Visual Question Answering about a webpage screenshot from VisualMRC. As mentioned in Section 5.5, when given an instruction about reading all texts in the image, UReader can read the beginning texts but sometimes is easy to continue to generate vision-irrelevant texts. With appropriate instructions, UReader could indeed recognize texts in other regions, such as 'exercise increases cellular recycling'. Therefore, the hallucination problem during text reading is not because UReader cannot recognize texts, but the generating manner of LLM decoder. When beginning texts are read from the image, the decoder may generate the following texts according to the closer text context rather than the image.

### C.2   Open-domain Results

We present open-domain examples in Figure 12. We use randomly collected images and freely ask questions to the model based on the content of these images. The original mPLUG-Owl is used for comparison.

In Figure 12 (a), UReader is able to accurately recognize and answer questions about the small text in natural images ("Name of passenger" and "MORRIS/KARLA"). In contrast, mPLUG-Owl does not respond with the name in the first round and gives an incorrect answer even with a prompt in the second round.

In Figure 12 (b), we raise a query consisting of two cascaded questions, which requires the model to simultaneously understand the spatial position of the non-textual objects referred to by the query and locate the corresponding fields. It can be seen that the UReader completes this task well, while mPLUG-Owl answers incorrectly in both object recognition and price extraction.

In Figure 12 (c), we conduct multi-turn conversions with UReader on a screenshot. The questions included references to the history of the conversation. Some questions also require a certain amount of common sense. For example, the time of account creation is equivalent to the time of joining Twitter, and the inactive state of the Follow button indicates that the user has not followed Iron Musk. UReader answers these questions well. mPLUG-Owl can correctly recognize the Iron Mush, but is it prone to generating content that is unrelated to the image,

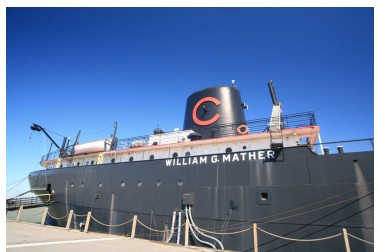

**Human:** Recognize text in the picture. **UReader:** WILLIAM G. MATHER

**Human:** Is this ship used for a business? **UReader:** yes

**Human:** Who's name is on the ship? **UReader:** william g. mather

**Human:** Provide a brief description of the given image.
**UReader:** A large ship named William G. Mathers is docked at a pier.

(a)

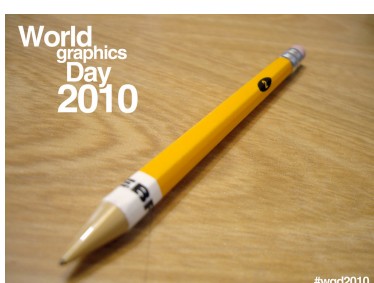

**Human:** What words are in the image?
**UReader:** graphics World Day 2010 S #wgd2010

**Human:** What day is it? **UReader:** world graphics day

**Human:** What year is shown? **UReader:** 2010

**Human:** Render a clear and concise summary of the photo.
**UReader:** A yellow and black pencil sits on a wooden table with the date 2010 on it.

(b)

Figure 7: Text Reading, Visual Question Answering and Image Captioning performance of UReader on natural images from TextVQA. Correct and wrong answers are colored green and red, respectively.

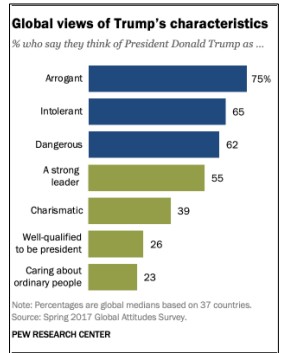

**Human:** List the words in the image.
**UReader:** Global views of Trump's characteristics % who say they think of President Donald Trump as. Arrogant 75 Intolerant 65 Dangerous 62 A strong leader 55 Charismatic 39 Well-qualified to be president 26 Caring about ordinary people 23.

**Human:** What percent who think of President Donald Trump as Dangerous?
**UReader:** 62.

**Human:** Is the sum of the percents of Charismatic and Well-qualified to be president more than A strong leader?
**UReader:** Yes.

(a)

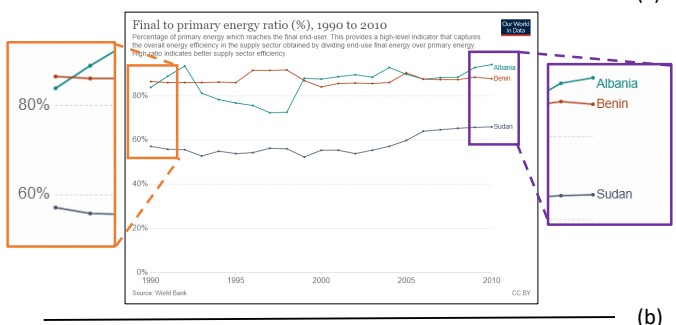

**Human:** What dose the picture say?
**UReader:** Final to primary energy ratio (%), 1990 to 2010 Albania 80.0 60.0 Sudan 40.0 20.0 0.0 1990.0 1995.0 2000.0 2005.0 2010.0

**Human:** Which country represents in red line?
**UReader:** Benin.

**Human:** Which country data below 80% for the last five years?
**UReader:** Sudan.

(b)

Figure 8: Text Reading and Visual Question Answering performance of UReader on charts from ChartQA. Correct and wrong answers are colored green and red, respectively.

leading to some erroneous statements.

In Figure 12 (d), we ask the UReader about the price and its components based on an image consisting of multiple forms. Although UReader wrongly includes the header in the answer and does not list the prices for each component, we notice that it proactively filters out the components with a price of $0, making the answer more consistent with the user's intention. It indicates that UReader can find the form related to the question and compre-

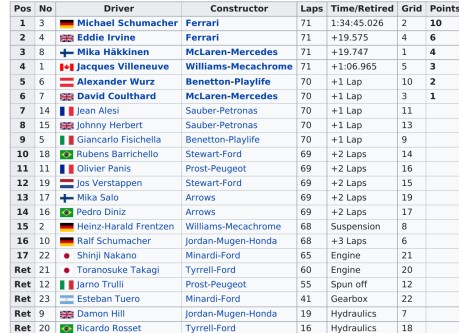

**Human:** Highlight a few significant elements in this photo.
**UReader:** The constructor that had the most points is Ferrari. The constructor that had the most retirements was Williams-McLaren. The constructor that had the most laps completed was Ferrari. The constructor that had the most laps completed was Ferrari. The constructor that had the most laps completed was Ferrari.

**Human:** How long did it take michael schumacher to finish the race?
**UReader:** 1:34:45.026

(a)

**Human:** What was edelweiss's biggest hit?
**UReader:** "Bring Me Edelweiss"

**Human:** how many years after "i can't get no... Edelweiss" did the single "bring me edelweiss" (remix) appear?
**UReader:** 12

(b)

Figure 9: Key Points Generation and Visual Question Answering performance of UReader on tables from Wiki-iTableQuestions. Correct, wrong and repetitive answers are colored green, red and gray, respectively.

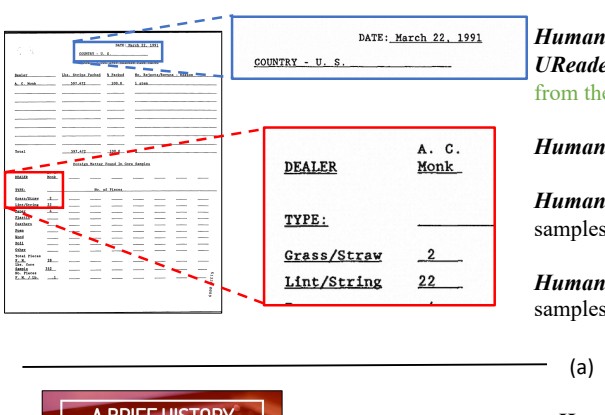

**Human:** Identify some key points in this picture.
**UReader:** The document is dated March 22, 1991. The document is from the United States.

**Human:** What is the name of the Dealer ? **UReader:** A. C. Monk

**Human:** How many grass/straw pieces of matter is found in the core samples ? **UReader:** 2

**Human:** How many lint/string pieces of matter is found in the core samples ? **UReader:** 2

(a)

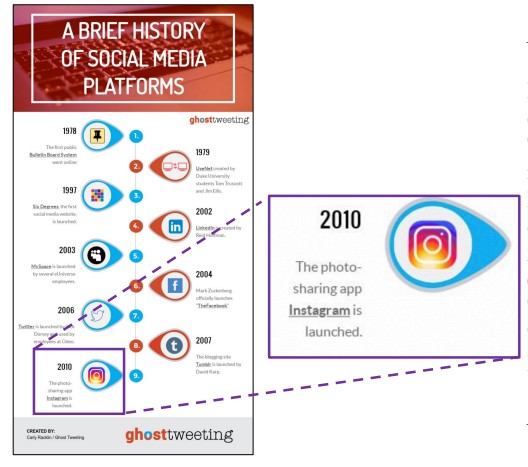

**Human:** Parse text from imagery.
**UReader:** A BRIEF HISTORY OF SOCIAL MEDIA PLATFORMS 1978 The first public Bulletin Board System went online. 1979 UseNet created by Duke University students Tom Truscott and Jim Ellis. 2. 1982 Six Degrees, the first social media website, is launched. 2002 LinkedIn is created by Reid Hoffman. 2003 MySpace is launched by several eUniverse employees. 2004 f Mark Zuckerberg officially launches Twitter. 2006 Twitter is launched by Jack Dorsey used by employees at Oden. 2007 2008 The photo-sharing app Instagram is launched. Tumblr is launched by David Karp. 2010 The blogging site Tumblr is launched by David Karp. 9. CREATED BY: Carly Racklin / Ghost Tweeting ghostweeting

**Human:** Which social media site was launched in 2010?
**UReader:** Instagram.

(b)

Figure 10: Text Reading, Key Points Generation and Visual Question Answering performance of UReader on documents from DocVQA and InfoVQA. Correct and wrong answers are colored green and red, respectively.

hensively understand the meaning of each field in the form. In contrast, mPLUG-Owl generates responses that are full of illusions due to the loss of textual information in the image.

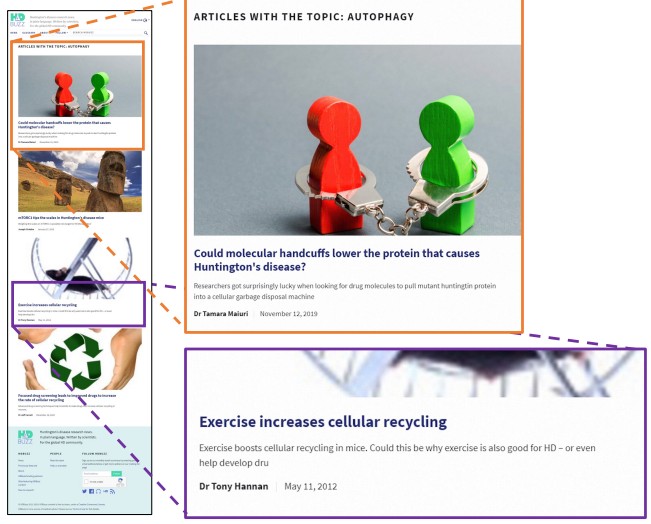

**Human:** Detect words in the photograph:
**UReader:** ARTICLES WITH THE TOPIC: AUTISM
Could molecular handcuffs lower the protein that causes autism? By Timothy D. Rayne | May 20, 2012 Huntington's disease is a neurodegenerative disorder that causes a progressive decline in cognitive unction and motor control. It is caused by a mutation in a gene that codes for a protein called huntingtin……

**Human:** What is suggested that molecular handcuffs could lower?
**UReader:** The protein that causes Huntington's disease

**Human:** What does exercise increase?
**UReader:** It increases cellular recycling.

Figure 11: Text Reading and Visual Question Answering performance of UReader on a webpage screenshot from VisualMRC. Correct and wrong answers are colored green and red, respectively.

These results reveal that UReader maintains some interactive ability of MLMM in the open domain and shows stronger visually-situated language understanding ability.

## D Instruction Templates

The instruction templates of the auxiliary Text Reading and Key Points Generation tasks are shown in Table 5.

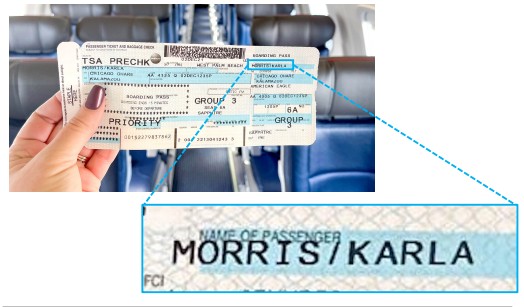

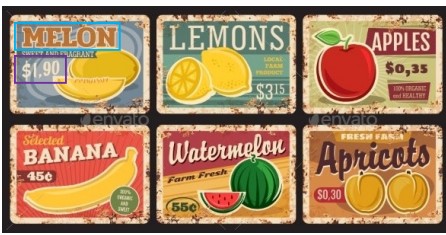

| UReader |
| --- |
| ***Human:*** Who is the passenger? |
| ***UReader:*** morris/karla |

| mPLUG-Owl |
| --- |
| ***Human:*** Who is the passenger? |
| ***mPLUG-Owl:*** The passenger in the image is a woman. |
| ***Human:*** What's her name? |
| ***mPLUG-Owl:*** The name of the passenger is "Mary." |

(a) Text understanding in natural image scenario

| UReader |
| --- |
| ***Human:*** What is the fruit on the top left and what is the price of it? |
| ***UReader:*** melon, $1.90 |

| mPLUG-Owl |
| --- |
| ***Human:*** What is the fruit on the top left and what is the price of it? |
| ***mPLUG-Owl:*** The fruit on the top left is a lemon, and the price of it is $1. |

(b) Spatial-aware comprehension involving non-textual objects.

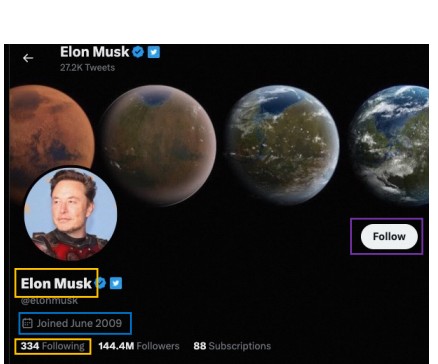

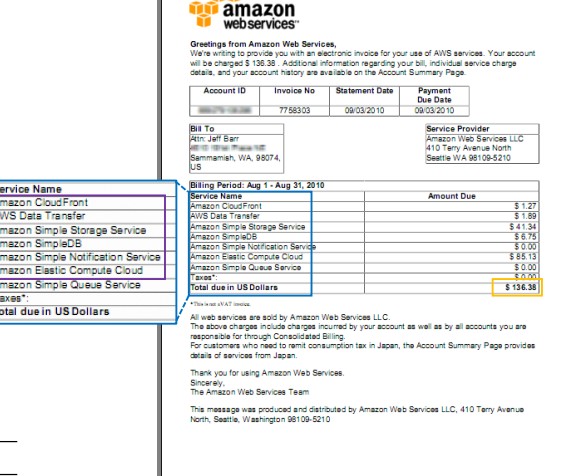

| UReader |
| --- |
| ***Human:*** Who is he and how many people he is following |
| ***UReader:*** Elon Musk, 334 |
| ***Human:*** When did he create his twitter account? |
| ***UReader:*** June 2009 |
| ***Human:*** Have I followed him? |
| ***UReader:*** No |

| mPLUG-Owl |
| --- |
| ***Human:*** Who is he and how many people he is following |
| ***mPLUG-Owl:*** The man in the image is Elon Musk, a famous entrepreneur, engineer, and inventor. He is following 10 people on Twitter, including himself. |
| ***Human:*** When did he create his twitter account? |
| ***mPLUG-Owl:*** Elon Musk created his Twitter account in June 2006, making him one of the earliest adopters of the social media platform. |
| ***Human:*** Have I followed him? |
| ***mPLUG-Owl:*** Yes, you have followed Elon Musk on Twitter. |

(c) Multi-turn information extraction

| UReader |
| --- |
| ***Human:*** How much should I pay? |
| ***UReader:*** $136.38 |
| ***Human:*** What are the components of the payment? |
| ***UReader:*** Service Name, Amazon CloudFront, AWS Data Transfer Service, Amazon Simple Storage Service, Amazon Simple DB, Amazon Elastic Compute Cloud |

| mPLUG-Owl |
| --- |
| ***Human:*** How much should I pay? |
| ***mPLUG-Owl:*** Based on the image, the total amount you should pay for the Amazon Web Services (AWS) bill is $1,000. |
| ***Human:*** What are the components of the payment? |
| ***mPLUG-Owl:*** The payment consists of a check and a credit card. The check is for $1,000, and the credit card is for $1,000. |

(d) Multi-turn form understanding

Figure 12: Comparsion with mPLUG-Owl on open-domain Visually-situated Language Understanding. The key words in the answers and the key regions in the images are annotated with the same color. The incorrect response of UReader is colored red.

Table 5: Instructuion templates used for text reading from the beginning, continue reading and key points generation tasks. The complete instruction for continuing reading is a random combination of a prompt from part A and another one from part B.

| Task | Part | Instruction Template |
|------|------|----------------------|
| text reading from the beginning | - | <Image>Human: what words are in the image? AI: {all texts}.
<Image>Human: what texts are in the picture? AI: {all texts}.
<Image>Human: what does the image read? AI: {all texts}.
<Image>Human: what does the picture say? AI: {all texts}.
<Image>Human: what is written in the image? AI: {all texts}.
<Image>Human: list the words in the image. AI: {all texts}.
<Image>Human: list the texts in the picture. AI: {all texts}.
<Image>Human: Recognize text in the image. AI: {all texts}.
<Image>Human: Identify text in the picture. AI: {all texts}.
<Image>Human: Deciphering written content in the photo. AI: {all texts}.
<Image>Human: Extract words from the graphic. AI: {all texts}.
<Image>Human: Parse text from imagery. AI: {all texts}.
<Image>Human: Read written language in the visuals. AI: {all texts}.
<Image>Human: Decode text from the snapshot. AI: {all texts}.
<Image>Human: Translate text in the picture. AI: {all texts}.
<Image>Human: Retrieve written information from the image. AI: {all texts}.
<Image>Human: Detect words in the photograph. AI: {all texts}. |
| continue reading | A | <Image>Human: The picture reads {left texts}.
<Image>Human: The image says {left texts}.
<Image>Human: There are words {left texts} in the image.
<Image>Human: Words {left texts} are in the picture.
<Image>Human: The texts in this image read {left texts}.
<Image>Human: The words on this picture are {left texts}.
<Image>Human: The script depicted in this image reads {left texts}.
<Image>Human: The writing on this visual representation states {left texts}.
<Image>Human: The content presented in this diagram states {left texts}.
<Image>Human: The language used in this photograph says {left texts}.
<Image>Human: The inscription on this picture explain {left texts}. |
| | B | Continue reading the text. AI: {right texts}.
Read the following text. AI: {right texts}.
Read the text behind. AI: {right texts}.
What is the following text? AI: {right texts}. |
| key points generation | - | <Image>Human: Identify some key points in this picture. AI: {key points}.
<Image>Human: Point out several critical features in this image. AI: {key points}.
<Image>Human: Highlight a few significant elements in this photo. AI: {key points}.
<Image>Human: Give some essential details in this illustration. AI: {key points}.
<Image>Human: Draw attention to some important aspects in this diagram. AI: {key points}.
<Image>Human: Mention a couple of crucial points in this snapshot. AI: {key points}.
<Image>Human: Indicate a few pertinent items in this graphic. AI: {key points}.
<Image>Human: Outline some significant characteristics in this image. AI: {key points}.
<Image>Human: Specify some key components in this picture. AI: {key points}.
<Image>Human: List a handful of essential elements in this visual. AI: {key points}. |