# OpenReview forum: "UReader: Universal OCR-free Visually-situated Language Understanding with Multimodal Large Language Model"
_EMNLP/2023/Conference — EMNLP 2023 Findings_

### Official Review · Reviewer_4B6Y · 2023-07-27

**Soundness:** 4

**Excitement:**

4: Strong: This paper deepens the understanding of some phenomenon or lowers the barriers to an existing research direction.

**Paper Topic And Main Contributions:**

This paper instruction tunes a pre-trained Multi-modal Large Language Model (MLLM) for OCR-free Visually-situated Language Understanding tasks. It converts an array of datasets into the instruction tuning format, tasks covering document, table, chart, natural images, webpage. Furthermore, the authors propose a Shape-Adaptive Cropping Module to better understand images with various resolutions, and Cropped Images Modeling for enhancing spatial relationship understanding. The final results show the effectiveness of this approach.

**Questions For The Authors:**

Please check the "Reasons To Reject" section.

**Reasons To Accept:**

1. Utilizing instruction tuning for OCR-free Visually-situated Language Understanding tasks makes sense

2. The performance is good

3. Ablation studies are exhaustive, showing the effects of different modules

4. The paper is well structured

**Reasons To Reject:**

1. If I understand correctly, the model is instruction tuned and evaluated on the same set of datasets. However, I believe instruction tuning can also help models to generalize to unseen datasets/tasks. I think it would be valuable to evaluate the model's zero-shot abilities on unseen datasets and unseen tasks, which would largely increase the impact of this work.

2. Lack of analyses of the performance on DocVQA. Also, DocVQA is one of the instruction tuning dataset, while authors said "although trained without document data" on line 477, is this inconsistent?

**Reproducibility:**

3: Could reproduce the results with some difficulty. The settings of parameters are underspecified or subjectively determined; the training/evaluation data are not widely available.

**Reviewer Confidence:**

4: Quite sure. I tried to check the important points carefully. It's unlikely, though conceivable, that I missed something that should affect my ratings.

---

> ### Author Rebuttal · Authors · 2023-08-29
>
> ### Q1: Model's zero-shot abilities on unseen datasets and unseen tasks.
>
> As mentioned In lines 470-480, we remove 4 document datasets from the training set and perform **zero-shot testing on DocVQA dataset** (table 2, r5).  Compared with a recent work utilizing LLM for zero-shot document image understanding by carefully designed layout-aware prompts [1], our zero-shot performance is much better **(46.2 vs 42.0)**.  Besides, we further test the **zero-shot performance of UReader on unseen dataset OCR-VQA**. With the same evaluation metrics, UReader outperforms mPLUG-Owl **(41.1 vs 28.6)** and a recent (Aug 19) work UniDoc [2] **(41.1 vs 34.5)** with the training of layout prediction. These two experiments show that **the zero-shot performance of our method on unseen domains is acceptable**.
>
> Besides, as shown in the figure 12, our model presents **multi-turn information extraction and multi-turn form understanding abilities**, which are unseen task formats during instruction finetuning. Therefore, our model shows some **generalization capabilities with unified instruction tuning**. We will further study how to strengthen this ability, such as developing chain-of-the-thought abilities in different domains.  Multi-turn information extraction.
>
> [1] Wang, Wenjin, et al. Layout and Task Aware Instruction Prompt for Zero-shot Document Image Question Answering. arXiv preprint arXiv:2306.00526 (2023).
>
> [2] Feng, H., Wang, Z., Tang, J., Lu, J., Zhou, W., Li, H., & Huang, C. (2023). UniDoc: A Universal Large Multimodal Model for Simultaneous Text Detection, Recognition, Spotting and Understanding. arXiv preprint arXiv:2308.11592.
>
> ### Q2:  Explain "although trained without document data" on line 477. The performance analyses of DocVQA.
>
> As mentioned In lines 470-472, The r5 in table 2 represents removing 4 document datasets from the training data. Line 477-480 refer to the  r5 rather than the UReader, so this is consistent.
>
> UReader underperforms ocr-free baselines on DocVQA because of two main factors:
>
> 1. Donut performs the pretraining on large-scale document dataset IIT-CDIP (11M document images), which is the same domain as DocVQA. But UReader is just instruction finetuned on ensembled datasets (less than 0.5M assorted images). Training with more document images brings better performance.
> 2. The pretraining task of Pix2Struct is to predict the HTML dom tree of a masked web screenshot, which requires the model to fully understand the layout information of the image. The pretraining on layout understanding leads to improved performance on DocVQA.
>
> The conclusion can also be substantiated by the observations on the other two datasets (i.e., InfoVQA and KLC) included in the document domain as previous work [1]. For the InfoVQA dataset, the image is poster style and the layout is not as important as DocVQA but the relationship between text and vision objects matters more, like natural image and chart image. As for the KLC dataset, ocr-free models are only fed with the first page (always the cover of a report) , where the layout is much simpler than DocVQA and DeepForm. Therefore, UReader can outperform baselines on these two document datasets.
>
> In summary, compared with ocr-free model Donut and  Pix2Struct, due to the pretrianing of MLMM on open-domain datasets, UReader is better at understanding cross-modality relationships in the image but weaker at comprehending text layout information without large-scale document pretraining and specific layout understanding tasks.
>
> In the future, we will study how to add as less extra document images as possible and design an auxiliary layout prediction task to strengthen the layout comprehension ability.
>
> [1] Tang, Z., Yang, Z., Wang, G., Fang, Y., Liu, Y., Zhu, C., ... & Bansal, M. (2023). Unifying vision, text, and layout for universal document processing. In Proceedings of the IEEE/CVF Conference on Computer Vision and Pattern Recognition (pp. 19254-19264).

---

### Official Review · Reviewer_4gdp · 2023-08-05

**Soundness:** 2

**Excitement:**

2: Mediocre: This paper makes marginal contributions (vs non-contemporaneous work), so I would rather not see it in the conference.

**Paper Topic And Main Contributions:**

This paper presents UReader, which is built upon a multimodal large language model and is capable of OCR ability.
UReader is tuned on a wide range of visually-situated language understanding datasets including documents, tables, charts, natural images, and webpage screenshots.
The paper also designs a shape-adaptive cropping module to utilize the frozen low-resolution vision encoder for processing high-resolution images.

**Reasons To Accept:**

The paper achieves promising experimental results and the method to process high-resolution images is interesting.

**Reasons To Reject:**

1. My main concern is the limited contribution of the paper. UReader can be seen as a multi-task fine-tuning on a wide range of visually-situated language understanding datasets such as DocVQA and InfographicsVQA. The more interesting part for me is the method to process high-resolution images.
2. In Table 1, UReader also performs fine-tuning (multi-task fine-tuning) on downstream tasks.
3. The train parameters of UReader are much smaller than other methods. But UReader uses CLIP-Large and LLaMA-7B as initialization, the total parameters of UReader are much larger than Pix2Struct.

**Reproducibility:**

4: Could mostly reproduce the results, but there may be some variation because of sample variance or minor variations in their interpretation of the protocol or method.

**Reviewer Confidence:**

3: Pretty sure, but there's a chance I missed something. Although I have a good feel for this area in general, I did not carefully check the paper's details, e.g., the math, experimental design, or novelty.

---

> ### Author Rebuttal · Authors · 2023-08-29
>
> ### Q1: Limited contribution of the paper.
>
> Thanks for your appreciation  for  the design of the shape-adaptive cropping module for processing high-resolution images.  For  achieving universal visually-situated language understanding, it's necessary to tackle various sizes of images. Existing MLLM could only process 224x224 or 448x448 images, which can not recognize texts in high-resolution document or website images. Therefore, how to leverage existing vision encoder process high-resolution images is the most difficult problem for developing a universal MLLM with low training cost. As one of the contributions in our work, the shape-adaptive cropping module is an elegant solution that avoids re-training a high-resolution encoder and could be adapted to other MLMM.
>
> Besides this model deisgn, there are another two contribution in our work:
>
> 1. As for the framework, we **first propose to develop universal visually-situated language understanding by only performing instruction tuning and dropping the high-cost domain-specific pretraining** used in previous methods like Donut and Pix2Struct. Under this framework, our method achieves OCR-free SOTA performance on 8/10 tasks and greatly reduces the training cost. Our model is trained on 20k steps and costs 16 A100 days, while Donut costs 192 A100 days and Pix2Struct-large is trained on 170k steps on 128 TPUS.
> 2. Besids unifying different tasks of different datasets, **we further design Text Reading and Key Points Generation tasks to strengthen models' text recognition and semantic understanding abilities**. As validated in our ablation study, these two auxiliary tasks significantly improve the performance, **e.g., +8.7 on DocVQA, +6.5 on ChartQA, and +16.7 on VisualMRC**.
>
> Considering these  aspects,  we argue that our contribution is three-fold and not just a multi-task finetuning.
>
> ### Q2: The usage of FT flag in Table 1.
>
> The "FT" flag represents independent finetuning on the specific downstream datasets. We will more clearly mention that the "FT" flag to avoid misunderstanding. Different from such a setting,  UReader is instruction-tuned with an ensembled dataset and trained with different tasks. With further specific finetune, UReader could also perform better on corresponding datasets. As shown in below, we further independently finetune UReader on 5 datasets, including DocVQA, WikiTableQuestion, ChartQA, TextVQA and VisualMRC, covering 5 categories. The 1st row is the raw performance of the unified UReader, The following rows illustrate the performance of the model after continuous fine-tuning on each dataset.  With further finetuning, there are indeed performance increases on corresponding datasets (e.g. +6.8 on VisualMRC, +2.6 on ChartQA, + 1.1 on DocVQA). However, this can cause performance decreases on other datasets. By utilizing existing MLLM, we aim to develop a single and unified model that could handle as many types of images as possible. Therefore, taking into account the balance of performance and versatile ability, we think the UReader with only unified instruction tuning is better. We will add this analysis to the revised version.
>
> ### Q3: The discussion about the model size of UReader.
>
> Trainable parameters listed in the table 1 aim to validate that to achieve universal visually-situated language understanding, leveraging an MLMM with our framework spends much less training cost than previous ocr-free models following the domain-specific pretrianing and finetuning framework. When comparing the whole model size,  more comprehensive abilities should be considered. For example, as shown in the FIgure 12 in the appendix, our single model presents multi-turn information extraction and form understanding abilities given diverse human instructions, which is not feasible for Donut or  Pix2Struct. Besides, due to that raw parameters of the LLM is frozen, our model is still able to process pure-text instruction as LLaMA, which is not also feasible for Donut or  Pix2Struct. We will explain more about the trainable parameters in the revised version.

---

### Official Review · Reviewer_v2FU · 2023-08-12

**Soundness:** 5

**Excitement:**

4: Strong: This paper deepens the understanding of some phenomenon or lowers the barriers to an existing research direction.

**Paper Topic And Main Contributions:**

The paper develops UReader, a vision language model developed for OCR-free visually situated language understanding.  It leverages an existing visual instruction tuning model (mPLUg-OWL) in their collection of unified instruction dataset covering general downstream task, text reading task, and key points generation. To deal with dense coverage of texts in the images, the authors propose a new shape-adpative cropping module that can easily adapt to various image shapes without domain-specific fine-tuning. Experiment results show that the model achieves state of the art ocr-free performance in 8 out of 10 visually situated language understanding task without finetuning. Ablation studies show interesting empirical findings: 1) auxiliary tasks of text reading and key points generation improve the model performance, 2) vision and language parts should be finetuned to adjust to their tasks, 3) shape adaptive cropping module with more number of crops, encoding cropped positional encoding, and adding resized global image  all bring improvements in performance.

**Questions For The Authors:**

- Given that UReader performs well in 8 out of 10 OCR-free tasks,  I am wondering what are reasons for not achieving leader scores in the other two tasks (DocVQA, DeepForm). What are some specific challenges encountered in these two tasks and what are some plans for improvements?

- Could you elaborate on the particular areas or factors where UReader falls short when compared to OCR pipeline models? What are the components that the authors thin UReader is missing to achieve human-level understanding?

-  Table 1: Shouldn't the model be considered as finetuned if the training data is included in the evaluation? What is the performance like if the model is then further finetuned on the evaluated dataset?

- The methodology for the text reading task involves randomly selecting a split position $p$ to process the input text in a conversational format. Could you provide some insights into the importance of this processing step? How would the model's performance be affected if all texts were treated solely as targets (i.e., $p=0$)?

**Reasons To Accept:**

- Achieves the state of the art results in 8 out of 10 tasks among OCR-free models with efficient training.
   - This is due to authors' optimization of training techniques, such as visual instruction tuning, LoRA, that enable fast training with small number of parameters to tune (16 A100 days with 86M) compared to prior work, e.g. 1.3B Pix2struct model that requires a batch size of 1024 on 128 TPUs.
  - Model training is reproducible compared to such large models proposed in prior work.
- Dense ablation studies in Table 2 show that each contribution of auxiliary tasks such as text reading (TR) and keypoints generation (KPG), and components in shape-adaptive cropping module bring improvement in performance.
- Qualitative examples are impressive in their fine-grained text recognition skills despite not relying on OCR.
- Model figures are clear to understand for future implementations and how the input images are processed for text recognition abilities.

**Reasons To Reject:**

- Work could be seen as limited novelty as the proposed UReader can be perceived primarily as an extension of the mPLUg-Owl model trained on instruction-formatted OCR dataset.

- While the proposed model demonstrates a solid performance against other OCR-free models, the authors do not thoroughly compare these results with those of existing OCR-pipeline models. It would be good to know how the model performs in the broader landscape of visually situated language understanding models.

- Relatively low performance on unseen data domains, e.g. documents, as demonstrated in Table 2 (r5 vs r10). This means that the model needs to be trained on the image domains to perform well, which is the major weakness of OCR free v.s. pipeline model that is not as affected by the image modalities.

**Reproducibility:**

4: Could mostly reproduce the results, but there may be some variation because of sample variance or minor variations in their interpretation of the protocol or method.

**Reviewer Confidence:**

3: Pretty sure, but there's a chance I missed something. Although I have a good feel for this area in general, I did not carefully check the paper's details, e.g., the math, experimental design, or novelty.

**Typos Grammar Style And Presentation Improvements:**

- L164, 170: "visuall" -> visually
- While the comparison to OCR free VL models is reasonable one, it would be helpful for readers to know the performance of models that use an OCR system to understand the gap between OCR free and reliant system.
- Table 2 can be broken down to multiple tables to clearly see the effects of the auxiliary tasks and cropping module. It is difficult to immediately understand the high-level story looking at this Table.

---

> ### Author Rebuttal · Authors · 2023-08-29
>
> ### Q1 (Reject 1): Novelty and Relation with mPLUG-Owl.
>
> UReader is proposed to extend a MLMM to universal visually-situated language understanding. In this work, we utilize but not limited to mPLUG-Owl. Rather than naively training a MLMM on a combined instruction-formatted dataset, the novelty of our model design lies in mainly two aspects:
>
> 1. In terms of model structure,  rather than training a high-resolution image encoder like Donut or Pix2Struct, we propose a shape-adaptive cropping module to fully leverage the original low-resolution (224x224) image encoder for processing high-resolution document images (e.g. 896x1120). As mentioned in Lines 481-488, this cropping module is critical for the universal visually-situated language understanding performance, e.g., dropping this cropping module (table 2, r6 vs r10) causes -43.4 on DocVQA, -35.1 on ChartQA, -64.3 on VisualMRC.
> 2. Besids unifying different tasks of different datasets, we further design Text Reading and Key Points Generation tasks to strengthen models' text recognition and semantic understanding abilities. As validated in our ablation study, these two auxiliary tasks significantly improve the performance, e.g., +8.7 on DocVQA, +6.5 on ChartQA, and +16.7 on VisualMRC.
>
> ### Q2 (Question 1): Dataset performance variation, challenges and improvement plans.
>
> UReader underperforms ocr-free baselines on DocVQA and DeepForm because of two main factors:
>
> 1. Donut performs the pretraining on large-scale document dataset IIT-CDIP (11M document images), which is the same domain as DocVQA and DeepForm. But UReader does no have a  pretraining process and is just instruction finetuned on ensembled datasets (less than 0.5M assorted images). Training with more document images brings better performance.
> 2. The pretraining task of Pix2struct is to predict the HTML dom tree of a masked web screenshot, which requires the model to fully understand the layout information of the image. But UReader is  trained to read texts from top to down, from left to right, which requires a weaker layout understanding ability. The pretraining on layout understanding also leads to improved performance on DocVQA.
>
> The conclusion can also be substantiated by the observations on the other two datasets (i.e., InfoVQA and KLC) included in the document domain as previous work [1]. For the InfoVQA dataset, the image is poster style and the layout is not as important as DocVQA and DeepForm but the relationship between text and vision objects matters more, like natural image and chart image. As for the KLC dataset, ocr-free models are only fed with the first page (always the cover of a report) , where the layout is much simpler than DocVQA and DeepForm. Therefore, UReader can outperform baselines on these two document datasets.
>
> In summary, compared with ocr-free model Donut and Pix2Struct, due to the pretrianing of MLMM on open-domain datasets, UReader is better at understanding cross-modality relationships in the image but weaker at comprehending text layout information without large-scale document pretraining and specific layout understanding tasks.
>
> In the future, we will study how to add as less extra document images as possible and design an auxiliary layout prediction task to strengthen the layout comprehension ability.
>
> [1] Tang, Z., Yang, Z., Wang, G., Fang, Y., Liu, Y., Zhu, C., ... & Bansal, M. (2023). Unifying vision, text, and layout for universal document processing. In Proceedings of the IEEE/CVF Conference on Computer Vision and Pattern Recognition (pp. 19254-19264).
>
> ### Q3 (Reject 2, Question 2): Compare with existing OCR-pipeline models. And the missing for human-level understanding.
>
> Thanks for your advice about comparing our model with SOTA pipelines for more in-depth analysis. We list the performance of SOTA pipeline models below.
>
> | |DocVQA|InfoVQA|DeepForm|KLC|WTQ|TabFact|ChartQA|TextVQA|TextCaps|VisualMRC|
> |-|-|-|-|-|-|-|-|-|-|-|
> |OCR-Pipline|84.7(UDOP)|47.4(UDOP)|85.5(UDOP)|82.8(UDOP)|47.2(UDOP)|72.9(UDOP)|70.5(DePlot)|56.3(PreSTU)|139.1 (PreSTU)|364.2(LayoutT5)|
> |UReader|	65.4|	42.2|	49.5|	32.8|	29.4|	67.6|	59.3|	57.6|	118.4|221.7|
>
> We can summarize from the results in the table that there are two distinct aspects. Firstly, our model achieves **comparable or slightly worse results** compared to the pipeline methods on **TextVQA, ChartQA, InfoVQA, TextCaps and TabFact**. Secondly, there is a **obvious gap** between our model and pipeline methods on **DocVQA, DeepForm, KLC, WTQ and VisualMRC**.
>
> For the first aspect, there are two reasons for the similarity performance:
>
> 1. **Modeling the diverse relationship between visual objects and text presents challenges for both pipeline-based methods and OCR-free methods**. TextVQA, TextCaps and InfoVQA requires the relation understanding between text and visual objects (i.e. logos, icons and common objects). ChartQA asks for trend comprehension of lines. Understanding such complex cross-modality relation is challenging for both ocr-free and pipeline methods.
> 2. **The simplicity of task formats can reduces performance gaps**. Tabfact is a simply binary classification task resulting the small performance gap.
>
> For this second aspect, the main performance gap appears in three categories of datasets: document, table, and webpage screenshot. The reasons are two folds:
>
> 1. **The gap in terms of text recognition and layout extraction**. In document, table and website, text is the dominant information source and the layout(e.g. row and column layout in table) is relatively uniformer than the chart and natural images. Therefore, with pre-extracted texts and layout information, it is more easy to understand the image. But for OCR-Free models, such as our UReader and Donut, it’s still challenging to fully recognize all texts.
> 2. **The gap in terms of modeling capacity on multi-page document input**. for multiple-page document datasets KLC (98% > 4 pages) and DeepForm (75% > 1 pages), OCR-Free models only input the first page and lose much information.
>
> Taking into account both two factors, UReader is not competenat on document, table and website domains.
>
> We will add this comparison and analysis in the revised version, In the future, to narrow this gap, we will further improve our model, including how to process multiple pages with a memory bank and how to strengthen the model’s layout comprehension with an auxiliary task. Besides, visually situated language understanding always requires math computation (e.g. table and chart understanding), to achieve human-level understanding, strengthening the math ability or incorporating a math component is necessary.
>
> ### Q4 (Reject 3): Relatively low performance on unseen data domains, e.g. documents, as demonstrated in Table 2 (r5 vs r10).
>
> Impressive zero-shot performance on unseen data domains is a challenging problem for not only OCR-free models but also pipeline models. SOTA pipeline models [1,2,3,5] all follow a pretrain and finetune framework. This is because the layout style and importance of different modalities vary a lot in different image types. For example, the row and column layout is crucial for table understanding, the vision semantic is important for poster-style documents, and the line trend is critical for chart understanding.
>
> Recently, a pipeline work [4] tried utilziing LLM for zero-shot document image understanding by designing layout-aware prompts. With the Alpaca (7B) as the LLM and carefully designed prompt, its pipeline zero-shot performance on DocVQA is 42.0, while ours ocr-free zero-shot performance is 46.2 (Table 2, r5). Besides, we further test the zero-shot performance of UReader on unseen dataset OCR-VQA, with the same evaluation metrics, UReader outperforms mPLUG-Owl (41.1 vs 28.6) and a recent (Aug 19) work UniDoc [6] (41.1 vs 34.5) with the training of layout prediction.
>
> This shows that under our framework, the zero-shot performance on unseen data domains is acceptable. We agree that such zero-shot performance is far from promising and will further study how to improve its zero-shot abilities in the future, such as introducing layout prediction tasks to strengthen the general vision understanding abilities.
>
> [1] Qiming Peng, Yinxu Pan, Wenjin Wang, Bin Luo, Zhenyu Zhang, Zhengjie Huang, Yuhui Cao, Weichong Yin, Yongfeng Chen, Yin Zhang, Shikun Feng, Yu Sun, Hao Tian, Hua Wu, and Haifeng Wang. 2022. ERNIE-Layout: Layout Knowledge Enhanced Pre-training for Visually-rich Document Understanding. In Findings of the Association for Computational Linguistics: EMNLP 2022. Association for Computational Linguistics, Abu Dhabi, United Arab Emirates, 3744–3756.
>
> [2] Chenliang Li, Bin Bi, Ming Yan, Wei Wang, Songfang Huang, Fei Huang, and Luo Si. 2021. StructuralLM: Structural Pre-training for Form Understanding. In ACL 2021.
>
> [3] Yupan Huang, Tengchao Lv, Lei Cui, Yutong Lu, and Furu Wei. 2022. LayoutLMv3: Pre-training for Document AI with Unified Text and Image Masking. In ACM MM 2022. arXiv:2204.08387
>
> [4] Wang, Wenjin, et al. Layout and Task Aware Instruction Prompt for Zero-shot Document Image Question Answering. arXiv preprint arXiv:2306.00526 (2023).
>
> [5] Tang, Z., Yang, Z., Wang, G., Fang, Y., Liu, Y., Zhu, C., ... & Bansal, M. (2023). Unifying vision, text, and layout for universal document processing. In Proceedings of the IEEE/CVF Conference on Computer Vision and Pattern Recognition (pp. 19254-19264).
>
> [6] Feng, H., Wang, Z., Tang, J., Lu, J., Zhou, W., Li, H., & Huang, C. (2023). UniDoc: A Universal Large Multimodal Model for Simultaneous Text Detection, Recognition, Spotting and Understanding. arXiv preprint arXiv:2308.11592.
>
> ### Q5 (Question 3): Table 1 should not use FT. And if the model is then further finetuned on the evaluated dataset?
>
> Thanks for your advice, we will more clearly mention that the "FT" flag represents specific finetuning on the downstream datasets to avoid misunderstanding. As you suggested, we further independently finetune UReader on 5 datasets, including DocVQA, WikiTableQuestion, ChartQA, TextVQA and VisualMRC, covering 5 categories. As shown below, the 1st row is the raw performance of the unified UReader. The following rows illustrate the performance of the model after continuous fine-tuning on each dataset.
>
> | |Document (DocVQA)|Chart (ChartQA)|Table (WikiTableQuestion)|Natural (TextVQA)|WebPage (VisualMRC)|
> |-|-|-|-|-|-|
> Instruction-tuned|65.4|59.3|29.4|57.6|221.7|
> Continuous fine-tuned on DocVQA (5.0k steps)|**66.5**|57.2|27.1|53.3|219.1|
> Continuous fine-tuned on ChartQA (2.0k steps)|58.1|**61.9**|24.5|49.1|194.8|
> Continuous fine-tuned on WikiTableQuestion (1.0k steps)|63.2|59.1|**29.7**|54.1|217.7|
> Continuous fine-tuned on TextVQA (1.1k steps)|58.6|57.0|26.3|**57.9**|202.4|
> Continuous fine-tuned on VisualMRC (1.8k steps)|63.54|59.25|27.63|54.76|**228.5**|
>
> With further finetuning, there are indeed performance increases on corresponding datasets (e.g. +6.8 on VisualMRC, +2.6 on ChartQA, + 1.1 on DocVQA). However, this can cause performance decreases on other datasets. By utilizing existing MLLM, we aim to develop a single and unified model that could handle as many types of images as possible. Therefore, taking into account the balance of performance and versatile ability, we think the UReader with only unified instruction tuning is better. We will add this analysis to the revised version.
>
> ### Q6 (Question 4): Insights into the importance of this processing step (text reading task involves randomly selecting a split position).
>
> As discussed in lines 315-318, we have noticed that the model's ability to predict text based on longer contexts is weaker compared to shorter contexts. During the inference process, the model often overlooks visual information and produces words inconsistent to the image when generating longer sequences. Therefore, we increase the training ratio for predicting text with longer contexts by randomly selecting starting points.
>
> For instance, when employing the teaching-forcing method on the sequence $T=[t_0,t_1,t_2,t_3,t_4]$, we calculate the loss value for predictions $P(t_i|ctx_{<i})\, i \in [1,2,3,4]$. If we specifically choose i=3 as the starting point, we build an extra training example and compute the loss value for predictions $P(t_i|ctx_{<i})\, i \in [3,4]$. As a result, loss values are calculated twice for $t_3$ and $t_4$ in this scenario.
>
> This approach can aid the model in gaining a more comprehensive understanding of the image.

---

### Meta-Review · Area_Chair_obfC · 2023-09-26

**Recommendation:** 4

**Metareview:**

The paper proposes a new approach to multimodal NLU using multimodal large language models. Two reviewers see a lot of potential in the paper, one is more reserved, but their reviews are less substantiated. The authors addressed several of reviewers’ concerns.

---

### Decision · Program_Chairs · 2023-10-07

**Decision:**

Accept-Findings

**Comment:**

The paper proposes a new approach to multimodal NLU using multimodal large language models. Two reviewers see a lot of potential in the paper, one is more reserved, but their reviews are less substantiated. The authors addressed several of reviewers’ concerns.